# Attention as Implicit Structural Inference

**Ryan Singh**
School of Engineering and Informatics,
University of Sussex.
rs773@sussex.ac.uk

**Christopher L. Buckley**
School of Engineering and Informatics,
University of Sussex.
VERSES AI Research Lab,
Los Angeles, CA, USA.

## Abstract

Attention mechanisms play a crucial role in cognitive systems by allowing them to flexibly allocate cognitive resources. Transformers, in particular, have become a dominant architecture in machine learning, with attention as their central innovation. However, the underlying intuition and formalism of attention in Transformers is based on ideas of keys and queries in database management systems. In this work, we pursue a structural inference perspective, building upon, and bringing together, previous theoretical descriptions of attention such as; Gaussian Mixture Models, alignment mechanisms and Hopfield Networks. Specifically, we demonstrate that attention can be viewed as inference over an implicitly defined set of possible adjacency structures in a graphical model, revealing the generality of such a mechanism. This perspective unifies different attentional architectures in machine learning and suggests potential modifications and generalizations of attention. Here we investigate two and demonstrate their behaviour on explanatory toy problems: (a) extending the value function to incorporate more nodes of a graphical model yielding a mechanism with a bias toward attending multiple tokens; (b) introducing a geometric prior (with conjugate hyper-prior) over the adjacency structures producing a mechanism which dynamically scales the context window depending on input. Moreover, by describing a link between structural inference and precision-regulation in Predictive Coding Networks, we discuss how this framework can bridge the gap between attentional mechanisms in machine learning and Bayesian conceptions of attention in Neuroscience. We hope by providing a new lens on attention architectures our work can guide the development of new and improved attentional mechanisms.

## 1   Introduction

Designing neural network architectures with favourable inductive biases lies behind many recent successes in Deep Learning. The Transformer, and in particular the attention mechanism has allowed language models to achieve human like generation abilities previously thought impossible [1, 2]. The success of the attention mechanism as a domain agnostic architecture has prompted adoption across a diverse range of tasks beyond language modelling, notably reaching state-of-the-art performance in visual reasoning and segmentation tasks [3, 4].

This depth and breadth of success indicates the attention mechanism expresses a useful computational primitive. Recent work has shown interesting theoretical links to kernel methods [5, 6, 7], Hopfield networks [8], and Gaussian mixture models [9, 10, 11, 12, 13], however a formal understanding that captures the generality of this computation remains outstanding. In this paper, we show the attention mechanism can naturally be described as inference on the structure of a graphical model, agreeing with observations that transformers are able to flexibly choose between models based on context [14, 15]. This Bayesian perspective complements previous theory [16, 8, 12], adding new

37th Conference on Neural Information Processing Systems (NeurIPS 2023).

methods for reasoning about inductive biases and the functional role of attention variables. Further, understanding the core computation as inference permits a unified description of multiple attention mechanisms in the literature as well as narrowing the explanatory gap to ideas in neuroscience.

This paper proceeds in three parts: First in Sec.3, we show that 'soft' attention mechanisms (e.g. self-attention, cross-attention, graph attention, which we call *transformer attention* hereafter) can be understood as taking an expectation over possible connectivity structures, providing an interesting link between softmax-based attention and marginal likelihood. Second in Sec.4, we extend the inference over connectivity to a Bayesian setting which, in turn, provides a theoretical grounding for iterative attention mechanisms (slot-attention and block-slot attention) [17, 18, 19], Modern Continuous Hopfield Networks [8] and Predictive Coding Networks. Finally in Sec.5, we leverage the generality of this description in order to design new mechanisms with predictable properties.

Intuitively, the attention matrix can be seen as the posterior distribution over edges $E$ in a graph, $\mathcal{G} = (K \cup Q, E)$ consisting of a set of query and key nodes $Q, K$ each of dimension $d$. Where the full mechanism computes an expectation of a function defined on the graph $V : \mathcal{G} \to \mathbb{R}^{d \times |\mathcal{G}|}$ with respect to this posterior.

$$
Attention(Q, K, V) = softmax(\overbrace{\frac{QW_Q W_K^T K^T}{\sqrt{d}}}^{p(E \mid Q, K)}) W_V K
$$
$$
= \mathbb{E}_{p(E|Q,K)}[V]
$$

Crucially, when $\mathcal{G}$ is seen as a graphical model, the posterior over edges becomes an inference about dependency structure and the functional form becomes natural. This formalism provides an alternate Bayesian theoretical framing within which to understand attention models, shifting the explanation from one centred around retrieval to one that is fundamentally concerned with in-context inference of probabilistic relationships (including retrieval). Within this framework different attention architectures can be described by considering different implicit probabilistic models, by making these explicit we hope to support more effective analysis and the development of new architectures.

## 2   Related Work

A key benefit of the perspective outlined here is to tie together different approaches taken in the literature. Specifically, structural variables can be seen as the alignment variables discussed in previous Bayesian descriptions [16, 20, 21], on the other hand Gaussian Mixture Models (GMMs) can be seen as a specific instance of the framework developed here. This description maintains the explanatory power of GMMs by constraining the alignment variables to be the edges of an implicit graphical model, while offering the increased flexibility of alignment approaches to describe multiple forms of attention.

**Latent alignment and Bayesian Attention,** several attempts have been made to combine the benefits of soft (differentiability) and stochastic attention, often viewing attention as a probabilistic alignment problem. Most approaches proceed by sampling, e.g., using the REINFORCE estimator [20] or a $topK$ approximation [22]. Two notable exceptions are [16] which embeds an inference algorithm within the forward pass of a neural network, and [21] which employs the re-parameterisation trick for the alignment variables. In this work, rather than treating attention weights as an independent learning problem, we aim to provide a parsimonious implicit model that would give rise to the attention weights. Additionally showing that 'soft' attention weights arise naturally in variational inference from either collapsed variational inference or a mean-field approximation.

**Relationship to Gaussian mixture model,** previous works that have taken a probabilistic perspective on the attention mechanism note the connection to inference in a gaussian mixture model [11, 10, 12, 13]. Indeed [23] directly show the connection between the Hopfield energy and the variational free energy of a Gaussian mixture model. Although Gaussian mixture models, a special case of the framework we present here, are enough to explain cross attention they do not capture slot or self-attention, obscuring the generality underlying attention mechanisms. In contrast, the description presented here extends to structural inductive biases beyond what can be expressed in a Gaussian mixture model, additionally offering a route to describing the whole transformer block.

**Attention as bi-level optimisation,** mapping feed-forward architecture to a minimisation step on a related energy function has been called unfolded optimisation [24]. Taking this perspective can lead to insights about the inductive biases involved for each architecture. It has been shown that the cross-attention mechanism can be viewed as an optimisation step on the energy function of a form of Hopfield Network [8], providing a link between attention and associative memory. while [25] extend this view to account for self-attention. Our framework distinguishes Hopfield attention, which does not allow an arbitrary value matrix, from transformer attention. Although there remains a strong theoretical connection, we interpret the Hopfield Energy as an instance of variational free energy, aligning more closely with iterative attention mechanisms such as slot-attention.

## 3 Transformer Attention

### 3.1 Attention as Expectation

We begin by demonstrating transformer attention can be seen as calculating an expectation over graph structures. Specifically, let $x = (x_1, .., x_n)$ be observed input variables, $\phi$ be some set of discrete latent variables representing edges in a graphical model of $x$ given by $p(x \mid \phi)$, and $y$ a variable we need to predict. Our goal is to find $\mathbb{E}_{y|x}[y]$, however the graph structure $\phi$ is unobserved so we calculate the marginal likelihood.

$$\mathbb{E}_{y|x}[y] = \sum_\phi p(\phi \mid x) \mathbb{E}_{y|x,\phi}[y]$$

Importantly, the softmax function is a natural representation for the posterior,

$$p(\phi \mid x) = \frac{p(x, \phi)}{\sum_\phi p(x, \phi)} = softmax(\ln p(x, \phi))$$

in order to expose the link to transformer attention, let the model of $y$ given the graph $(x, \phi)$ be parameterised by a function $\mathbb{E}_{y|x,\phi}[y] = v(x, \phi)$.

$$\mathbb{E}_{y|x}[y] = \sum_\phi softmax(\ln p(x, \phi)) v(x, \phi) = \mathbb{E}_{\phi|x}[v(x, \phi)] \tag{1}$$

In general, transformer attention can be seen as weighting $v(x, \phi)$ by the posterior distribution $p(\phi \mid x)$ over different graph structures. We show Eq.1 is exactly the equation underlying self and cross-attention by presenting the specific generative models corresponding to them. In this description the latent variables $\phi$ are identified as edges between observed variables $x$ (keys and queries) in a pairwise Markov Random Field, parameterised by matrices $W_K$ and $W_Q$, while the function $v$ is parameterised by $W_V$.

**Pairwise Markov Random Fields** are a natural tool for modelling the dependencies of random variables, with prominent examples including Ising models (Boltzmann Machine) and multivariate Gaussians. While typically defined given a known structure, the problem of inferring the latent graph is commonly called structural inference.

Formally, given a set of random variables $X = (X_v)_{v \in V}$ with probability distribution $[p]$ and a graph $G = (V, E)$. The variables form a pairwise Markov Random Field (pMRF) [26] with respect to $G$ if the joint density function $P(X = x) = p(x)$ factorises as follows

$$p(x) = \frac{1}{Z} \exp\left( \sum_{v \in V} \psi_v + \sum_{e \in E} \psi_e \right)$$

where $Z$ is the partition function $\psi_v(x_v)$ and $\psi_e = \psi_{u,v}(x_u, x_v)$ are known as the node and edge potentials respectively. Bayesian structural inference also requires a structural prior $p(\phi)$ over the space of possible adjacency structures, $\phi \in \Phi$, of the underlying graph.

**Factorisation,** without constraints this space grows exponentially in the number of nodes ($2^{|V|}$ possible graphs leading to intractable softmax calculations), all the models we explore here implicitly assume a factorised prior[1]. We briefly remark that Eq.1 respects factorisation of $[p]$ in the following

---

[1]Additionally placing zero probability mass on much of the space, for example disconnected graphs.

sense; if the distribution admits a factorisation (a partition of the space of graphs $\Phi = \prod_i \Phi_i$) with respect to the latent variables $p(x, \phi) = \prod_i e^{f_i(x, \phi_i)}$ where $\phi_i \in \Phi_i$, and the value function distributes over the same partition of edges $v(x, \phi) = \sum_i v_i(x, \phi_i)$ then each of the factors can be marginalised independently:

$$\mathbb{E}_{\phi|x}[v(x, \phi)] = \sum_i \mathbb{E}_{\phi_i|x}[v_i] \tag{2}$$

To recover cross-attention and self-attention we need to specify the structural prior, potential functions and a value function. (In order to ease notation, when $\Phi_i$ is a set of edges involving a common node $x_i$, such that $\phi_i = (x_i, x_j)$ represents a single edge, we use the notation $\phi_i = [j]$, suppressing the shared index.)

### 3.2 Cross Attention and Self Attention

We first define the model that gives rise to cross-attention:

- Key nodes $K = (x_1, .., x_n)$ and query nodes $Q = (x'_1, ..., x'_m)$
- Structural prior $p(\phi) = \prod_{i=1}^{m} p(\phi_i)$, where $\Phi_i = \{(x_1, x'_i), .., (x_n, x'_i)\}$ is the set of edges involving $x'_i$ and $\phi_i \sim Uniform(\Phi_i)$ such that each query node is uniformly likely to connect to each key node.
- Edge potentials $\psi(x_j, x'_i) = x'^T_i W^T_Q W_K x_j$, in effect measuring the similarity of $x_j$ and $x'_i$ in a projected space.
- Value functions $v_i(x, \phi_i = [j]) = W_V x_j$, a linear transformation applied to the node at the start of the edge $\phi_i$.

Taking the expectation with respect to the posterior in each of the factors defined in Eq.2 gives the standard cross-attention mechanism,

$$\mathbb{E}_{p(\phi_i|Q, K)}[v_i] = \sum_j softmax_j(x'^T_i W^T_Q W_K x_j) W_V x_j$$

If the key nodes are in fact the same as the query nodes and the prior is instead over a directed graph we recover self-attention (A.8.1).

## 4 Iterative Attention

We continue by extending attention to a latent variable setting, where not all the nodes are observed. In essence applying the attention trick, i.e., a marginalisation of structural variables, to a variational free energy (Evidence Lower Bound). This allows us to recover models such as slot attention [17] and block-slot attention [18]. These mechanisms utilise an EM-like procedure using the current estimation of latent variables to infer the structure and then using the inferred structure to improve estimation of latent variables. Interestingly, Modern Continuous Hopfield Networks fit within this paradigm rather than the one discussed in Sec.3; collapsed variational inference produces an identical energy function to the one proposed by Ramsauer et al. [8].

### 4.1 Collapsed Inference

We present a version of collapsed variational inference [27], where the collapsed variables $\phi$ are again structural, showing how this results in a Bayesian attention mechanism. In contrast to the previous section, we have a set of (non-structural) latent variables $z$. The goal is to infer $z$ given the observed variables, $x$, and a latent variable model $p(x, z, \phi)$. Collapsed inference proceeds by marginalising out the extraneous latent variables $\phi$ [27]:

$$p(x, z) = \sum_\phi p(x, z, \phi) \tag{3}$$

We define a gaussian recognition density $q(z) \sim N(z; \mu, \Sigma)$ and optimise the variational free energy $\mathcal{F}(\lambda) = \mathbb{E}_q[\ln q_\lambda(z) - \ln p(x, z)]$ with respect to the parameters, $\lambda = (\mu, \Sigma)$, of this distribution. Application of Laplace's method yields approximate derivatives of the variational

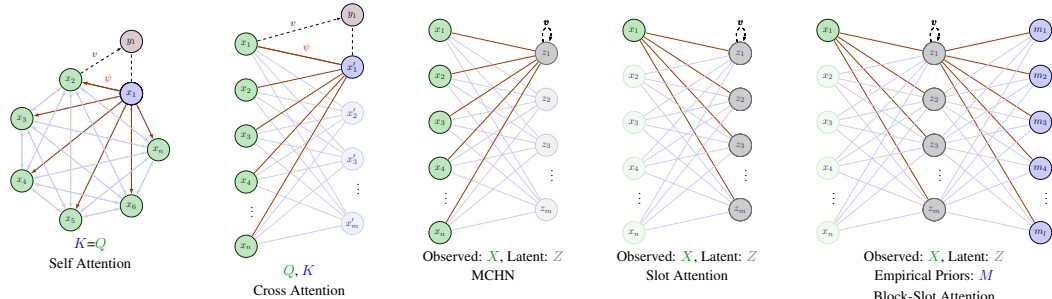

Figure 1: Comparison of models involved in different attention mechanisms. In each case, the highlighted edges indicate $\Phi_i$ the support of the uniform prior over $\phi_i$. Attention proceeds by calculating a posterior over these edges, given the current state of the nodes, before using this inference to calculate an expectation of the value function $v$. For iterative attention mechanisms the value function can be identified as the gradient of a variational free energy, in contrast, transformer attention uses a learnable function.

free energy $\nabla_\mu \mathcal{F} \approx -\nabla_\mu \ln p(x, \mu)$ and $\nabla_\Sigma \mathcal{F} \approx -\nabla_\mu^2 \ln p(x, \mu)$, here we focus on the first order terms [2]. Substituting in Eq.3:

$$\nabla_\mu \mathcal{F} \approx -\nabla_\mu \ln \sum_\phi p(x, \mu, \phi) \tag{4}$$

$$= -\frac{1}{\sum_\phi p(x, \mu, \phi)} \sum_\phi \nabla_\mu p(x, \mu, \phi) \tag{5}$$

In order to make the link to attention, we employ the log-derivative trick, substituting $p(\cdot) = e^{\ln p(\cdot)}$ and re-express Eq.5 in two ways:

$$= -\sum_\phi softmax_\phi(\ln p(x, \mu, \phi))\nabla_\mu \ln p(x, \mu, \phi) \tag{6}$$

$$= \mathbb{E}_{\phi|x,\mu}[-\nabla_\mu \ln p(x, \mu, \phi)] \tag{7}$$

The first form reveals the softmax which is ubiquitous in all attention models. The second, suggests the variational update should be evaluated as the expectation of the typical variational gradient (the term within the square brackets) with respect to the posterior over the parameters represented by the random variable $\phi$. In other words, iterative attention is exactly transformer attention applied iteratively where the value function is the variational free energy gradient. We derive updates for a general pMRF before again recovering (iterative) attention models in the literature by specifying particular distributions.

**Free Energy of a marginalised pMRF,** recall the factorised pMRF, $p(x, \phi) = \frac{1}{Z} \prod_i e^{f_i(x, \phi_i)}$. Again, independence properties simplify the calculation, the marginalisation can be expressed as a product of local marginals, $\sum_\phi p(x, \phi) = \frac{1}{Z} \prod_i \sum_{\phi_i} e^{f_i(x, \phi_i)}$. Returning to the inference setting, the nodes are partitioned into observed nodes, $x$, and variational parameters $\mu$. Hence the (approximate) collapsed variational free energy Eq.5, can be expressed as, $F(x, \mu) = -\sum_i \ln \sum_{\phi_i} e^{f_i(x, \mu, \phi_i)} + C$ and it's derivative:

$$\frac{\partial F}{\partial \mu_j} = -\sum_i \sum_{\phi_i} softmax(f_i)\frac{\partial f_i}{\partial \mu_j}$$

Finally, we follow [8] in using the Convex-Concave Procedure (CCCP) to derive a simple fixed point equation which necessarily reduces the free energy.

**Quadratic Potentials and the Convex Concave Procedure,** assuming the node potentials are quadratic $\psi(x_i) = -\frac{1}{2}x_i^2$ and the edge potentials have the form $\psi(x_i, x_j) = x_i W x_j$, and define $\tilde{f}_i = \sum_{e \in \Phi_i} \psi_e$ . Consider the following fixed point equation,

$$\mu_j^* = \sum_i \sum_{\phi_i} softmax(\tilde{f}_i)\frac{\partial \tilde{f}_i}{\partial \mu_j} \tag{8}$$

---

[2]As the first order terms are independent of the second order ones, see A.7.1 for details.

since (under mild conditions) node potentials are convex and edge potentials are concave (A.7.1.1), we can invoke the CCCP [28] to show this fixed point equation descends on the energy $F(x, \mu_j^*) \leq F(x, \mu_j)$ with equality if and only if $\mu_j^*$ is a stationary point of $F$. We follow Sec.3 in specifying specific structural priors and potential functions that recover different iterative attention mechanisms.

## 4.2 Modern Continuous Hopfield Network

Let the observed, or memory, nodes $x = (x_1, .., x_n)$ and latent nodes $z = (z_1, .., z_m)$ have the following structural prior $p(\phi) = \prod_{i=1}^{m} p(\phi_i)$, where $\phi_i \sim Uniform\{(x_1, z_i), .., (x_n, z_i)\}$, meaning each latent node is uniformly likely to connect to a memory node. Define edge potentials $\psi(x_j, z_i) = z_i^T x_j$. Application of Eq.8:

$$\mu_i^* = \sum_j softmax_j(\mu_i^T x_j)x_j$$

When $\mu_i$ is initialised to some query $\xi$ the system the fixed point update is given by $\mu_i^*(\xi) = \mathbb{E}_{\phi_i|x,\xi}[x_{[j]}]$. If the patterns $x$ are well separated, $\mu_i^*(\xi) \approx x_{j'}$, where $x_{j'}$ is the closest vector and hence can be used as an associative memory.

## 4.3 Slot Attention

Slot attention [17] is an object centric learning module centred around an iterative attention mechanism. Here we show this is a simple adjustment of the prior beliefs on our edge set. With edge potentials of the form $\psi(x_j, z_i) = z_i^T W_Q^T W_K x_j$, replace the prior over edges with $p(\phi) = \prod_{j=1}^{n} p(\phi_j)$, $\phi_j \sim Uniform\{(x_j, z_1), .., (x_j, z_m)\}$. Notice, in comparison to MCHN, the prior over edges is swapped, each observed node is uniformly likely to connect to a latent node, in turn altering the index of the softmax.

$$\mu_i^* = \sum_j softmax_i(\mu_i^T W_Q^T W_K x_j)W_Q^T W_K x_j$$

while the original slot attention employed an RNN to aid the basic update shown here, the important feature is that the softmax is taken over the 'slots'. This forces competition between slots to account for the observed variables, creating object centric representations.

## 4.4 Predictive Coding Networks

Predictive Coding Networks (PCN) have emerged as an influential theory in Computational Neuroscience [29, 30, 31]. Building on theories of perception as inference and the Bayesian brain, PCNs perform approximate Bayesian inference by minimising a variational free energy of a graphical model, where incoming sensory data are used as observations. Typical implementations use a hierarchical model with Gaussian conditionals, resulting in a local prediction error minimising scheme. The minimisation happens on two distinct time-scales, which can be seen as E-step and M-steps on the variational free energy: a (fast) inference phase encoded by neural activity corresponding to perception and a (slow) learning phase associated with synaptic plasticity. Gradient descent on the free energy gives the inference dynamics for a particular neuron $\mu_i$, [32]

$$\frac{\partial \mathcal{F}}{\partial \mu_i} = -\sum_{\phi^-} k_\phi \epsilon_\phi + \sum_{\phi^+} k_\phi \epsilon_\phi w_\phi$$

Where $\epsilon$ are prediction errors, $w$ represent synaptic strength, $k$ are node specific precisions representing uncertainty in the generative model and $\phi^-, \phi^+$ represent pre-synaptic and post-synaptic terminals resectively. Applying a uniform prior over the incoming synapses results in a slightly modified dynamics,

$$\frac{\partial \mathcal{F}}{\partial \mu_i} = -\sum_{\phi^-} softmax(-\epsilon_\phi{}^2)k_\phi \epsilon_\phi + \sum_{\phi^+} softmax(-\epsilon_\phi{}^2)k_\phi \epsilon_\phi w_\phi$$

where the softmax function induces a normalisation across prediction errors received by a neuron. This dovetails with theories of attention as normalisation in Psychology and Neuroscience [33, 34, 35]. In contrast previous predictive coding based theories of attention have focused on the precision terms, $k$, due to their ability to up and down regulate the impact of prediction errors [36, 37, 38]. Here we

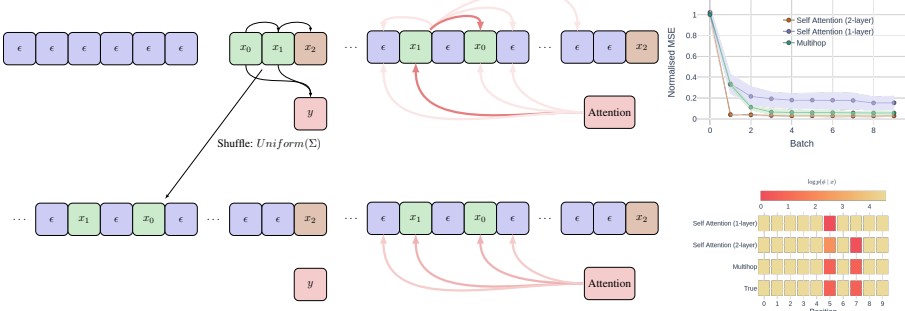

Figure 2: Multihop Attention: (left) Graphical description of the toy problem, $x_2$ is generated causally from $x_1$ and $x_0$, which are used to generate $y$. (centre) Comparison of the attention employed by Multihop which takes two steps on the attention graph (top) contrasted with Self Attention (bottom). Multihop Attention has the correct bias to learn the task approaching the performance of two-layer Self Attention, while a single layer of Self Attention is unable (top right). Empirically examining the attention weights, Multihop Attention is able to balance attention across two positions, while self-attention favours a single position.

see the softmax terms play a functionally equivalent role to precision variables, inheriting their ability to account for bottom-up and top-down attention, while exhibiting the fast winner-takes-all dynamics that are associated with cognitive attention.

# 5 New Designs

By identifying the attention mechanism in terms of an implicit probabilistic model, we can review and modify the underlying modelling assumptions in a principled manner to design new attention mechanisms. Recall transformer attention can be written as the marginal probability $p(y \mid x) = \sum_\phi p(\phi \mid x)\mathbb{E}_{y|x,\phi}[y]$, the specific mechanism is therefore informed by three pieces of data: (a) the value function $p(y \mid x, \phi)$, (b) the likelihood $p(x \mid \phi)$ and (c) the prior $p(\phi)$. Here, we explore modifying (a) and (c) and show they can exhibit favourable biases on toy problems.

## 5.1 Multi-hop Attention

Our description makes it clear that the value function employed by transformer attention can be extended to any function over the graph. For example, consider the calculation of $\mathbb{E}_{y|x,\phi}[y_i]$ in transformer attention, a linear transformation is applied to the most likely neighbour, $x_j$, of $x_i$. A natural extension is to include a two-hop neighbourhood, additionally using the most likely neighbour $x_k$ of $x_j$. The attention mechanism then takes a different form $\mathbb{E}_{p(\phi_j|\phi_i)p(\phi_i|x)}[V(x_{\phi_i} + x_{\phi_j})] = (P_\phi + P_\phi^2)VX$, where $P_\phi$ is the typical attention matrix. While containing the same number of parameters as a single-layer of transformer attention, for some datasets two-hop attention should be able to approximate the behaviour of two-layers of transformer attention.

**Task Setup** We simulate a simple dataset that has this property using the following data generation process: Initialise a projection matrix $W_y \in \mathbb{R}^{d \times 1}$ and a relationship matrix $W_r \in \mathbb{R}^{d \times d}$. $X$ is then generated causally, using the relationship $x_{i+1} = W_r x_i + N(0, \sigma)$ to generate $x_0$, $x_1$ and $x_2$, while the remaining nodes are sampled from the noise distribution $N(0, \sigma)$. Finally, the target $y$ is generated from the history of $x_2$, $y = W_y(x_1 + x_0)$ and the nodes of $X$ are shuffled. Importantly $W_r$ is designed to be low rank, such that performance on the task requires paying attention to both $x_1$ and $x_0$, Figure 2.

## 5.2 Expanding Attention

One major limitation of transformer attention is the reliance on a fixed context window. From one direction, a small context window does not represent long range relationships, on the other hand a large window does an unnecessary amount of computation when modelling a short range relationship. By replacing the uniform prior with a geometric distribution $p(\phi \mid q) \sim Geo(q)$,

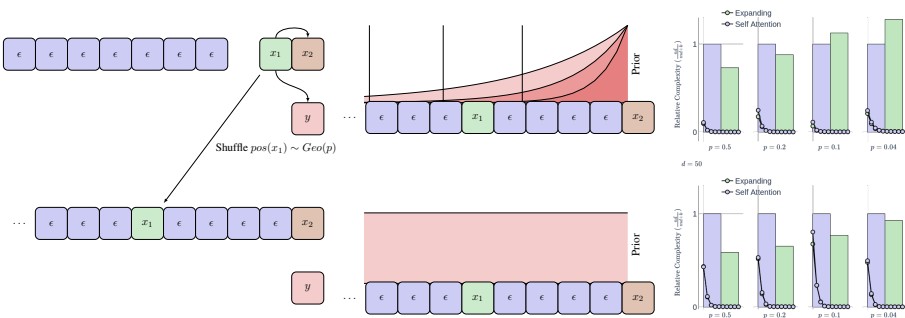

Figure 3: Expanding Attention: (left) Graphical description of the toy problem, $x_2$ and $y$ are generated from $x_1$ which is shuffled with a (exponentially decaying) recency bias. (centre) Comparison of the geometric prior, with different shades of red representing the iterative refinements during inference, used by Expanding and uniform prior used by Self Attention. (right) The relative number of operations used by Expanding Attention is beneficial when either the recency bias ($1/p$) or the number of feature dimensions ($d$) is large, training curves (overlaid) across each of these settings remained roughly equivalent.

and a conjugate hyper-prior $p(q) \sim Beta(\alpha, \beta)$ we derive a mechanism that dynamically scales depending on input. We use a (truncated) mean-field variational inference procedure [39] to iteratively approximate $p(\phi \mid x)$ using the updates: 1. $q_t \leftarrow \frac{\beta_t}{\alpha_t + \beta_t}$, 2. $p_t = p(\phi \mid x, q_t)$, 3. $\alpha_{t+1} \leftarrow \alpha_t + 1$, $\beta_{t+1} \leftarrow \beta_t + \sum_{<H(q_t)} i(p_t)_i$. Where $\alpha$ and $\beta$ are hyperparameters determining the strength of the prior and $H$ is the truncation horizon. Since attention dot products can be cached and reused for each calculation of step 2. the iterative procedure is computationally cheap.

The attention mechanism has asymptotic time complexity $O(n^2 d)$ where $n$ is the size of the size of the context window and $d$ is dimension over which the inner product is computed. In comparison, expanding attention $O(n(md + k))$ where $m$ is the size of the window at convergence, and $k$ is the number of steps to converge. If, as is typical, $d$ is large such that $d >> k$ the time complexity of expanding attention should be favourable.

**Task Setup** Input and target sequence are generated similarly to above (without $x_0$). Here $x_1$ is moved away from $x_2$ according to a draw from a geometric distribution, Figure 3.

## 6 Discussion

### 6.1 The Full Transformer Block

Transformer attention is typically combined with residual connections and a feedforward network, both of which have been shown important in preventing 'token collapse'. Here we briefly touch upon how these features might relate to the framework presented here.

**Feedforward layer,** it has previously been noticed the feedforward component can also be understood as a key-value memory where the memories are stored as persistent weights [40, 41]. This is due to the observation $ff(x) = W_2\sigma(W_1 x)$ is equivalent to attention when the non-linearity $\sigma$ is a softmax, although a ReLU is typically used. We speculate the framework presented here could be extended explain this discrepancy, intuitively the ReLU relates to an edge prior that fully factorises into binary variables.

**Residual connections** have been shown to encourage iterative inference [42]. This raises the possibility transformer attention, rather than having an arbitrary transformation $v$ as presented in Sec.3, is in fact approximately implementing the iterative inference of Sec.4 through a form of iterative amortised inference [43]. The view that the transformer is performing iterative refinement is additionally supported by empirical studies of early-decoding [44].

**Temperature and positional encodings,** both positional encodings and the temperature scaling can be seen as adjustments to the prior edge probability. In the case of relative positional encodings, by breaking the permutation invariance of the prior (A.8.2). While the temperature may be understood

in terms of tempered (or generalised) Bayesian inference [45], adjusting the strength of the prior relative to the likelihood.

## 6.2 Limitations

The connection to structural inference presented here is limited to the attention computation of a single transformer head, an interesting future direction would be to investigate whether multiple layers and multiple heads typically used in a transformer can also be interpreted within this framework. Additionally, the extension to iterative inference employed a crude approximation to the variational free energy, arguably destroying the favourable properties of Bayesian methods. Suggesting the possibility of creating iterative attention mechanisms with alternative inference schemes, possibly producing more robust mechanisms.

## 6.3 Conclusion

In this paper, we presented a probabilistic description of the attention mechanism, formulating attention as structural inference within a probabilistic model. This approach builds upon previous research that connects cross attention to inference in a Gaussian Mixture Model. By considering the discrete inference step in a Gaussian Mixture Model as inference on marginalised structural variables, we bridge the gap with alignment-focused descriptions. This framework naturally extends to self-attention, graph attention, and iterative mechanisms, such as Hopfield Networks. We hope this work will contribute to a more unified understanding of the functional advantages and disadvantages brought by Transformers.

Furthermore, we argue that viewing Transformers from a structural inference perspective provides different insights into their central mechanism. Typically, optimising structure is considered a learning problem, changing on a relatively slow timescale compared to inference. However, understanding Transformers as fast structural inference suggests that their remarkable success stems from their ability to change effective connectivity on the same timescale as inference. This general idea can potentially be applied to various architectures and systems. For instance, Transformers employ relatively simple switches in connectivity compared to the complex dynamics observed in the brain [46]. Exploring inference over more intricate structural distributions, such as connectivity motifs or modules in network architecture, could offer artificial systems even more flexible control of resources.

## Acknowledgements

This work was supported by The Leverhulme Trust through the be.AI Doctoral Scholarship Programme in biomimetic embodied AI. Additional thanks to Alec Tschantz, Tomasso Salvatori, Miguel Aguilera and Tomasz Korbak for their invaluable feedback and discussions.

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

Table 1: Different attention modules

| Name | Graph ($G$) | Prior ($p(\phi)$) | Potentials ($\psi$) | Value $v(x,\phi)$ |
|------|-------------|-------------------|---------------------|-------------------|
| Cross Attention | Key nodes $K$, query nodes $Q$ | Uniform | $x_i'^T W_Q^T W_K x_j$ | $V x_j$ |
| Self Attention | K = Q, directed edges | Uniform | $x_i^T W_Q^T W_K x_j$ | $V x_j$ |
| Graph Attention, Sparse Attention | K = Q, directed edges | Uniform (restricted) | $x_i^T W_Q^T W_K x_j$ | $V x_j$ |
| Relative Positional Encodings | K = Q, directed edges | Categorical | $x_i^T W_Q^T W_K x_j$ | $V x_j$ |
| Absolute Positional Encodings | K = Q | Uniform | $\tilde{x}_i^T W_Q^T W_K \tilde{x}_j$ $\tilde{x}_i = x_i + e_i$ | $V x_j$ |
| Classification Layer | NN output $f_\theta(X)$, classes $y$ | Uniform | $f_\theta(X)_i^T y_j$ | $y_j$ |
| MCHN | Observed nodes $X$, latent nodes $Z$ | Uniform (observed) | $z_i^T W_Q^T W_K x_j$ | $\frac{\partial F}{\partial z}$ |
| Slot Attention | Observed nodes $X$ latent nodes $Z$ | Uniform (latent) | $z_i^T W_Q^T W_K x_j$ | $\frac{\partial F}{\partial z}$ |
| Block-Slot Attention | Observed nodes $X$, latent nodes $Z$, memory nodes $M$ | Uniform (latent) | $z_i^T W_Q^T W_K x_j$, $m_k^T W_Q^T W_K z_i$ | $\frac{\partial F}{\partial z}$ |
| PCN | Observed nodes $X$, multiple layers of latent nodes $\{Z^{(l)}\}_{l \leq L}$ | Uniform (latent) | $z_i^T W_Q^T W_K x_j$ | $\frac{\partial F}{\partial z}$ |
| Multihop Attention | K = Q, directed edges | Uniform | $x_i^T W_Q^T W_K x_j$ | $V x_j + V x_k$ |
| Expanding Attention | K = Q, directed edges | Geometric x Beta | $x_i^T W_Q^T W_K x_j$ | $V x_j$ |

# 7 Appendix

Here we include some more detailed derivations of claims made in the paper, and list the hyperparameters used for the experiments.

## 7.1 Iterative Attention

In this section we provide a more detailed treatment of the Laplace approximation, and provide proper justification for invoking the CCCP. For both, the following lemma is useful:

**Lemma 7.1.** *The function* $\ln p(x) = \ln \sum_\phi p(x,\phi) = \ln \sum_\phi \exp E_\phi(x)$ *has derivatives (i)* $\frac{\partial}{\partial x} \ln p(x) = \mathbb{E}_{\phi|x}[\frac{\partial}{\partial x} E_\phi]$ *and (ii)* $\frac{\partial^2}{\partial x^2} \ln p(x) = Var_{\phi|x}[\frac{\partial}{\partial x} E_\phi] + \mathbb{E}_{\phi|x}[\frac{\partial^2}{\partial x^2} E_\phi]$

*Proof.* Let $E = (E_\phi)$ the vector of possible energies, and $p = (p_\phi) = (p(\phi \mid x))_\phi$ the vector of conditional probabilities. Consider $\ln p(\phi \mid x)$ written in canonical form,

$$\ln p(\phi \mid x) = \langle E_\phi(x), \mathbb{1}_\phi \rangle - A[E_\phi(x)] + h(\phi)$$

Where $A[E(x)] = \ln Z(E)$ is the cumulant generating function. By well known properties of the cumulant: $\frac{\partial A}{\partial E_i} = p(\phi = i \mid x) = p_i$. Hence by the chain rule for partial derivatives, $\frac{\partial A}{\partial x} = \sum_\phi p(\phi \mid x) \frac{\partial}{\partial x} E_\phi$, which is (i).

To find the second derivative we apply again the chain-rule $\frac{d}{dt} f(g(t)) = f''(g(t))g'(t)^2 + f'(g(t))g''(t)$. Again by properties of the cumulant $\frac{\partial^2 A}{\partial E_i \partial E_j} = Cov(\mathbb{1}_i, \mathbb{1}_j) = [diag(p) - p^T p]_{i,j} =$

$\mathbb{V}_{i,j}$. Hence the second derivative is

$$\frac{\partial^2 A}{\partial^2 x} = \frac{\partial E}{\partial x}^T \mathbb{V} \frac{\partial E}{\partial x} + \mathbb{E}[\frac{\partial^2 E_\phi}{\partial x^2}] \tag{9}$$

$\square$

**Second order Laplace Approximation** With these derivatives in hand we can calculate the second order laplace approximation of the free energy $\mathcal{F} = \mathbb{E}_q[\ln q_\lambda(z) - \ln p(x, z)]$.

$$\mathcal{F} \approx \mathbb{E}_q[\ln p(\mu, x) + \frac{\partial}{\partial z} \ln p(\mu, x)^T (z - \mu) + (z - \mu)^T \frac{\partial^2}{\partial z^2} \ln p(\mu, x)(z - \mu)] + H[q]$$

$$\approx \ln p(\mu, x) + tr(\Sigma_q^{-1} Var_{\phi|\mu,x}[\frac{\partial}{\partial z} E_\phi]) + tr(\Sigma_q^{-1} \mathbb{E}_{\phi|\mu,x}[\frac{\partial^2}{\partial z^2} E_\phi]) + \frac{1}{2} log \mid \Sigma_q \mid + C$$

We can see optimising the first order variational parameter in this approximation is independent of $\Sigma_q$, hence we can first find $\mu$ and the fill in our uncertainty $\Sigma_q = \frac{\partial^2}{\partial z^2} \ln p(\mu^*, x) = Var_{\phi|\mu,x}[\frac{\partial}{\partial z} E_\phi] + \mathbb{E}_{\phi|\mu,x}[\frac{\partial^2}{\partial z^2} E_\phi]$. Finding this uncertainty can be costly in the general case where the hessian of $E$ is not analytically available.

As alluded to in the paper, iterative attention mechanisms can also be viewed as an alternating maximisation procedure, which may provide a route to more general inference schemes:

**As Alternating Minimisation** Collapsed Inference can also be seen as co-ordinate wise variational inference [27]. Consider the family of distributions $Q = \{q(z; \lambda)q(\phi \mid z)\}$, where $q(z; \lambda)$ is parameterised, however $q(\phi)$ is unconstrained.

$$\mathcal{F} = \min_{q \in Q} \mathbb{E}_q[\ln q(z, \phi) - \ln p(x, z, \phi)]$$

$$= \min_{q \in Q} \mathbb{E}_{q(z)}[\mathbb{E}_{q(\phi)}[\ln q(\phi) - \ln p(x, \phi \mid z)] + \ln q(z) - \ln p(z)]$$

The inner expectation is maximised for $q(\phi) = p(\phi \mid x, z)$ and the inner expectation evaluates to $-\ln p(x \mid z)$ which recovers the marginalised objective

$$\min_{q \in Q} \mathbb{E}_{q(z)}[q(z) - \ln \sum_\phi p(x, z, \phi)]$$

This motivates an alternate derivation of iterative attention as structural inference which is less reliant on the Laplace approximation; Consider optimising over the variational family $Q = \{q(z; \lambda)q(\phi)\}$ coordinate wise:

$$\ln q_{t+1}(\phi) = \mathbb{E}_{q_t(z; \lambda_t)}[\ln p(\phi \mid x, z)] + C$$

$$\lambda_{t+1} = arg \min_\lambda \mathbb{E}_{q_t(\phi)}[\mathbb{E}_{q(z; \lambda)}[\ln q(z) - \ln p(x, z \mid \phi)]]$$

$$= arg \min_\lambda \mathbb{E}_{q_t(\phi)}[\mathcal{F}_\phi]$$

In the case of quadratic potentials, $q_{t+1}(\phi) = p(\phi \mid x, \lambda_t)$, hence the combined update step can be written

$$arg \min_\lambda \mathbb{E}_{p(\phi|x, \lambda_t)}[\mathcal{F}_\phi(\lambda)]$$

Each step necessarily reduces the free energy of the mean-field approximation, so this process converges. This derivation is independent of which approximation or estimation is used to minimise the typical variational free energy.

### 7.1.1 Convexity details for the CCCP

Given a pairwise pMRF with quadratic potentials $\psi(x_i) = -\frac{1}{2}x_i^2$ and the edge potentials have the form $\psi(x_i, x_j) = x_i W x_j$ and W p.s.d., s.t. $\ln p(x, \phi) = -\frac{1}{2}\sum_{v \in \mathcal{G}} x_v^2 + \ln \sum_\phi \exp g_\phi(x)$, where $g_\phi(x) = \sum_{e \in \phi} \psi_e$. We need the following lemma to apply the CCCP:

**Lemma 7.2.** $\ln \sum_\phi \exp g_\phi(x)$ *is convex in* $x$.

*Proof.* We reapply Lemma.7.1, with $E_\phi = g_\phi(x)$, hence $\frac{\partial^2}{\partial x^2} \ln \sum_\phi \exp g_\phi(x) = Var_{\phi|x}[\frac{\partial}{\partial x} g_\phi] + \mathbb{E}_{\phi|x}[\frac{\partial^2}{\partial x^2} g_\phi]$. The first matrix is a variance, so p.s.d. The second term $\mathbb{E}_{\phi|x}[\sum_{e \in \phi} \frac{\partial^2}{\partial x^2} \psi_e]$ is a convex sum of p.s.d matrices. Hence both terms are p.s.d, implying $\ln \sum_\phi \exp g_\phi(x)$ is indeed convex. $\square$

## 7.2 PCN Detailed Derivation

Here we go through the derivations for the equations presented in section 4.4. PCNs typically assume a hierarchical model with gaussian residuals:

$$z_0 \sim N(\hat{\mu}_0, \Sigma_0)$$
$$z_{i+1} \mid z_i \sim N(f_i(z_i; \theta_i), \Sigma_i)$$
$$y \mid z_N \sim N(f_N(z_N; \theta_N), \Sigma_N)$$

Under these conditions, a delta approximation of the variational free energy is given by:

$$\mathcal{F}[p, q] = \mathbb{E}_{q(z;\mu)}[-\ln p(y, z)] + H[q]$$

$$\mathcal{F}(\mu, \theta) \approx \sum_{l=0}^{N} \Sigma_l^{-1} \epsilon_l^2$$

Where $\epsilon_l = (\mu_{l+1} - f_l(\mu_l; \theta_l))^2$. The inference phase involves adjusting the parameters, $\mu$ in the direction of the gradient of $\mathcal{F}$, which for a given layer is:

$$\frac{\partial \mathcal{F}}{\partial \mu_l} = \Sigma_{l-1}^{-1} \epsilon_{l-1} - \Sigma_l^{-1} \epsilon_l f'(\mu_l) \tag{10}$$

Here, for ease of comparison, we consider the case where the link functions are linear, $f_i(\cdot) = W_i(\cdot)$ and further the precision matrices are diagonal $\Sigma_i^{-1} = diag(k_i)$. Under these conditions we can write the derivative component-wise as sums of errors over incoming and outgoing edges :

$$\left(\frac{\partial \mathcal{F}}{\partial \mu_l}\right)_i = -\sum_{\phi^-} k_\phi \epsilon_\phi + \sum_{\phi^+} k_\phi \epsilon_\phi w_\phi$$

Where $\phi^-, \phi^+$ represent the set of incoming and outgoing edges respectively, and we redefine $\epsilon_\phi = (\mu_i - \mu_j w_{ij})$ for an edge $\phi = (z_i, z_j)$ and $k_\phi = K(z_j)$ the precision associated with the node at the terminus of $\phi$.

Now if we instead assume a uniform prior over incoming edges, or concretely;

$$z_0 \sim N(\hat{\mu}_0, \Sigma_0)$$
$$\phi_l^i \sim Uniform(\{(z_{l+1}^i, z_l^0), (z_{l+1}^i, z_l^1), ...\})$$
$$z_{l+1}^i \mid z_l, \phi_l^i \sim N(w_l^{ij} z_l^{\phi_l^i}, 1/k_l^i)$$
$$y \mid z_N \sim N(f_N(z_N; \theta_N), \Sigma_N)$$

The system becomes a pMRF with edge potentials given by the prediction errors, recall applying Eq.4:

$$\frac{\partial F}{\partial \mu_j} = -\sum_i \sum_{\phi_i} softmax(f_i(x, \mu, \phi_i)) \frac{\partial f_i}{\partial \mu_j}$$

Here for a node in a given layer, it participates in one $\Phi_{l-1}^j$ and all the $\Phi_{l+1}^k$ from the layer above, where every $f_i(x, \mu, \phi_i)$ here is a squared prediction error corresponding to the given edge $e_l^{ij} =$

$k_l^{ij}(z_l^i - w_l^{ij} z_{l-1}^j)^2$, hence:

$$\frac{\partial F}{\partial \mu_j} = -\sum_{i \in \Phi_{l-1}^j} softmax_i(-(\epsilon_{l-1}^{ij})^2)\epsilon_{l-1}^{ij} k_j$$

$$+ \sum_{k \in [l]} \sum_{i' \in \Phi_l^k} softmax_{i'}(-(\epsilon_l^{i'k})^2)\epsilon_l^{i'k} w_l^{i'k} \mathbb{1}(i' = j)$$

$$\frac{\partial F}{\partial \mu_j} = -\sum_{i \in \Phi_{l-1}^j} softmax_i(-(\epsilon_{l-1}^{ij})^2)\epsilon_{l-1}^{ij}$$

$$+ \sum_{k \in [l]} softmax_{i'}(-(\epsilon_l^{i'k})^2)\epsilon_l^{i'k} w_l^{i'k}$$

Here incoming signals (nodes $i$) compete through the softmax, whilst the outgoing signal competes with other outgoing signals from nodes (nodes $i'$) in the same layer for representation in the next layer (nodes $k$), see block-slot attention diagram for intuition. By abuse of notation (reindexing edges as $\phi$)

$$\frac{\partial \mathcal{F}}{\partial \mu_i} = -\sum_{\phi^-} softmax(-\epsilon_\phi^2)k_\phi\epsilon_\phi + \sum_{\phi^+} softmax(-\epsilon_\phi^2)k_\phi\epsilon_\phi w_\phi$$

While we derived these equations for individual units to draw an easy comparison to standard Predictive Coding, we note it is likely more useful to consider blocks of units competing with each other for representation, similar to multidimensional token representations in typical attention mechanisms. We also briefly note here, the Hammersley–Clifford theorem indicates a deeper duality between attention as mediated by precision matrices and as structural inference.

## 7.3 New Designs

**Multihop Derivation** $\mathbb{E}_{y|x,\phi}[y_i]$ in transformer attention, a linear transformation is applied to the most likely neighbour, $x_j$, of $x_i$. A natural extension is to include a two-hop neighbourhood, additionally using the most likely neighbour $x_k$ of $x_j$. Formally, the value function $v$ no longer neatly distributes over the partition $\Phi_i$, however the attention mechanism then takes a different form: $\mathbb{E}_{p(\phi_j|\phi_i)p(\phi_i|x)}[V(x_{\phi_i} + x_{\phi_j})] = (P_\phi + P_\phi^2)VX$. Where we use $\phi_{j(i)} = \phi_j$ to denote the edge set of the node at the end of $\phi_i$. To see this note:

$$\mathbb{E}_{p(\phi|x)}[V(x_{\phi_i} + x_{\phi_j})] = \sum_\phi \prod_k p(\phi_k \mid x)V(x_{\phi_i} + x_{\phi_j})$$

$$= \sum_\phi \prod_k p(\phi_k \mid x)V(x_{\phi_i} + x_{\phi_j})$$

$$= \sum_\phi \prod_k p(\phi_k \mid x)V x_{\phi_i} + \sum_\phi \prod_k p(\phi_k \mid x)V x_{\phi_j}$$

by independence properties

$$= \sum_{\phi_i} p(\phi_i \mid x)V x_{\phi_i} + \sum_{\phi_i, \phi_j} p(\phi_i \mid x)p(\phi_j \mid x)V x_{\phi_j}$$

Denoting the typical attention matrix, $P$, where $p_{ij} = p(\phi_i = [j] \mid x)$

$$= \sum_k \sum_j p_{jk}p_{ij}V x_k + \sum_j p_{ij}V x_j$$

$$= (P_\phi + P_\phi^2)VX$$

**Expanding Derivation** As in the main text, let $p(\phi \mid q) \sim Geo(q)$ and $p(q) \sim Beta(\alpha, \beta)$, such that we have the full model $p(x, \phi, q; \alpha, \beta) = p(x \mid \phi)p(\phi \mid q)p(q; \alpha, \beta)$. In order to find $p(\phi \mid x)$ we employ a truncated Mean Field Variational Bayes [39], assuming a factorisation $p_t(\phi, q) = p_t(\phi)p_t(q)$, and using the updates:

$$\ln p_{t+1}(\phi) = \mathbb{E}_{p_t(q)}[\ln p(x \mid \phi) + \ln p(\phi \mid q)] + C_1$$

$$\ln p_{t+1}(q) = \mathbb{E}_{p_t(\phi)}[\ln p(\phi \mid q) + \ln p(q; \alpha, \beta)] + C_2$$

By conjugacy the second equation simplifies to a simple update of the beta distribution
$$\implies p_{t+1}(q) = Beta(\alpha_{t+1}, \beta_{t+1})$$
$$\alpha_{t+1} = \alpha_t + 1$$
$$\beta_{t+1} = \beta_t + \mathbb{E}_{p_t(\phi)}[\phi]$$
While the second update can be seen as calculating the posterior given $q_t = \mathbb{E}_{p_t(q)}[q]$,
$$\ln p_{t+1}(\phi) = \ln p(x \mid \phi) + \mathbb{E}_{p_t(q)}[\ln p(\phi \mid q)] + C_2$$
$$= \ln p(x \mid \phi) + \phi \mathbb{E}_{p_t(q)}[\ln q] + C_2$$
$$= \ln p(\phi \mid x, q_t)$$
Finally, we use a truncation to approximate the infinite sum $\mathbb{E}_{p_t(\phi)}[\phi] = \sum_k p_t(\phi = k)k \approx \sum_{<H} p_t(\phi = k)k$. Where we set the horizon according to the current distribution of $q$. For example in our experiments we chose $H(q_t) = \ln 0.05/ \ln(1 - q_t)$ the truncation that would capture 95% of the probability mass of the prior.

# 8 Attention Variants

Here we briefly discuss some variants of attention that there wasn't space for in the paper.

## 8.1 Self Attention

- Nodes $K = Q = (x_1, .., x_n)$
- Structural prior, over a fully connected, directed graph $p(\overrightarrow{\phi}) = \prod_{i=1}^{n} p(\overrightarrow{\phi}_i)$, where $\overrightarrow{\Phi}_i = \{(x_1, x_i), .., (x_n, x_i)\}$ is the set of edges involving $x_i$ and $\overrightarrow{\phi}_i \sim Uniform(\overrightarrow{\Phi}_i)$, such that each node is uniformly likely to connect to every other node in a given direction.
- Edge potentials $\psi(x_j, x_i) = x_i^T W_Q^T W_K x_j$, in effect measuring the similarity of $x_j$ and $x_i'$ in a projected space.
- Value functions $v_i(x, \phi_i = [j]) = W_V x_j$, a linear transformation applied to the node at the start of the edge $\phi_i$.

Again, taking the posterior expectation in each of the factors defined in two Eq.2 gives the standard self-attention mechanism
$$\mathbb{E}_{p(\phi_i|Q)}[v_i] = \sum_j softmax_j(x_i^T W_Q^T W_K x_j) W_V x_j$$

## 8.2 Positional Encodings and Graph Neural Networks

In Table.1 we show that positional encodings and graph attention are naturally incorporated in this framework. Absolute positional encoding as suggested by Vaswani et al. [1] can be seen as modifying the edge potentials with a vector that depends on position, while relative position encodings can be seen as a categorical prior, where the prior depends on the relative distance between nodes. Graph and Sparse attention operate similarly to graph attention, except the uniform prior is restricted to edges in the provided graph, or according to predefined sparsity pattern.

**Relative Position Encodings** If the prior over edges is categorical i.e. $P(\phi_i = [j]) = p_{i,j}$, it can be fully specified by the matrix $(P)_{i,j} = p_{i,j}$. This leads to the modified attention update
$$\sum_j softmax_j(x_i Q^T K x_j + \ln p_{ij}) x_j$$
However this requires local parameters for each node $z_i$. A more natural prior assign a different probability to the relative distance of $i$ from $j$. This is achieved with $P = circ(p_1, p_2, .., p_n)$, where $circ$ is the circulant matrix of $(p)_{i \leq n}$. Due to properties of circulant matrices $\ln P_{ij}$ can be reparameterised with the hartley transform
$$\ln p_{i,j} = \sum_k \beta_k [cos(k\theta_{i,j}) + sin(k\theta_{i,j})] = \beta \cdot b^{(i,j)}$$

Where $b_k^{(i,j)} = cos(k\frac{i-j}{2\pi n}) + sin(k\frac{i-j}{2\pi n})$ can be thought of as a relative position encoding, and $\beta$ are parameters to be learnt.

### 8.2.1 Block-Slot Attention

Singh et al. [18] suggest combining an associative memory ability with an object-centric slot-like ability and provide an iterative scheme for doing so, alternating between slot-attention and hopfield updates. Our framework permits us to flexibly combine different attention mechanisms through different latent graph structures, allowing us to derive a version of block-slot attention.

In this setting we have three sets of variables $X$, the observations, $Z$ the latent variables to be inferred and $M$ which are parameters. Define the pairwise MRF $X = \{x_1, ..., x_n\}$, $Z = \{z_1, ..., z_m\}$ and $M = \{m_1, ..., m_l\}$ with a prior over edges $p(E) = \prod_{j=1}^m p(E_j) \prod_{k=1}^l p(\tilde{E}_k)$, $E_j \sim Uniform\{(x_j, z_1), .., (x_j, z_m)\}$, $\tilde{E}_k \sim Uniform\{(z_1, m_k), .., (z_m, m_k)\}$, with edge potentials between $X$ and $Z$ given by $\psi(x_j, z_i) = z_i Q^T K x_j$ and between $Z$ and $M$, $\psi(z_i, m_k) = z_i \cdot m_k$ applying (8) gives

$$\mu_i^* = \sum_j softmax_i(\mu_i Q^T K x_j) Q^T K x_j$$
$$+ \sum_k softmax_k(\mu_i \cdot m_k) m_k$$

In the original block-slot attention each slot $z_i$ is broken into blocks, where each block can access block-specific memories i.e. $z_i^{(b)}$ can has possible connections to memory nodes $\{m_k^{(b)}\}_{k \leq l}$. Allowing objects to be represented by slots which in turn disentangle features of each object in different blocks. We presented a single block version above, however it is easy to see that the update extends to the multiple block version applying (8) gives

$$\mu_i^* = \sum_j softmax_i(\mu_i Q^T K x_j) Q^T K x_j$$
$$+ \sum_{k,b} softmax_k(\mu_i^{(b)} \cdot m_k^{(b)}) m_k^{(b)}$$

## 9 Experimental Details

**Multihop**

Following the notation in the text; data generation parameters:

- Total number of tokens: 10
- Embedding dimension (dimension of each $x$):10
- Output dimension (dimension of $y$): 1
- $\sigma^2$ (autoregressive noise): 1
- Random matrix initialisation was performed with torch.rand

Training parameters (across all models):

- batch size:200
- number of batches: 10
- optimiser: ADAM
- learning rate: $1e - 3$
- Number of different random seeds: 10

Model: To make analysis easier, all models were prevented from self-attending to the final token.

**Expanding**

Following the notation in the text; data generation parameters:

- Total number of tokens: 50

- Embedding dimension(s) (dimension of each $x$):[10, 50]
- $p$ the parameter for generating a geometric shuffle:[0.5, .2, .1, .04]
- Output dimension (dimension of $y$): 1
- $\sigma^2$ (autoregressive noise): .1
- Random matrix initialisation was performed with torch.rand

Training parameters (across all models):

- batch size:1*
- number of batches: 10000
- optimiser: ADAM
- learning rate: $5e-4$

Model: To make analysis easier, all models were prevented from self-attending to the final token. For expanding attention the hyperparameters were set as $\alpha = .1$, $\beta = .9$ these were chosen to have a mean value at roughly a quarter of the (size 50) window.

*Training was performed with single samples, despite the iterative process being completely parallel (no shared state). Naive parallel implementation of expanding attention would encounter synchronisation locks, as the fastest samples wait for the longest ones to complete. In order to take full advantage of a dynamic window over a batch, intelligent asynchronous processing would be necessary.

---

**Algorithm 1** Attention

---

**Require:** $X, W_q, W_k, W_v$
$\quad Q \leftarrow W_q X$
$\quad K \leftarrow W_k X$
$\quad V \leftarrow W_v X$
$\quad A \leftarrow softmax(K^T Q)$
$\quad Y \leftarrow AV$

---

**Algorithm 2** Multihop

---

**Require:** $X, W_q, W_k, W_v, N$
$\quad Q \leftarrow W_q X$
$\quad K \leftarrow W_k X$
$\quad V \leftarrow W_v X$
$\quad A \leftarrow softmax(K^T Q)$
$\quad P \leftarrow \sum_{k<N} A^k$
$\quad Y \leftarrow PV$

---

**Algorithm 3** Expanding

---

**Require:** $X, x_q, W_q, W_k, W_v, \alpha, \beta, k = 0$
$\quad q \leftarrow W_q x_q$
$\quad K \leftarrow W_k X$
$\quad V \leftarrow W_v X$
$\quad$ **while** $k$ not converged **do**
$\quad\quad p = \alpha/(\alpha + \beta)$
$\quad\quad k \leftarrow \ln(.05)/\ln(1-p)$
$\quad\quad g_p[i] = -i/k$
$\quad\quad A \leftarrow softmax(q^T K[:k,:] + g_p)$
$\quad\quad \alpha \leftarrow \alpha + 1$
$\quad\quad \beta \leftarrow \beta + \sum_{i<k} A[i]i$
$\quad$ **end while**
$\quad y \leftarrow AV$

---

