# OpenReview forum: "Attention as Implicit Structural Inference"
_NeurIPS.cc/2023/Conference — NeurIPS 2023 poster_

### Official Review · Reviewer_14Rb · 2023-06-28

**Soundness:** 3 good
**Presentation:** 2 fair
**Contribution:** 4 excellent
**Rating:** 8
**Confidence:** 3

**Summary:**

The paper shows how attention mechanisms can be interpreted as expectation values over learnable graph connectivity structures given a structural prior; that is from a perspective of (structural) variational inference. The authors first demonstrate this link for cross- and self-attention heads, then proceed with interpreting iterative attention mechanisms as performing gradient descent on an approximate variational free energy. Finally, based on this perspective of variational structural inference, the authors propose two new attention mechanisms which allow for more complicated connectivity structures.

Please note that due to receiving 6 papers to review from NeurIPS alone, I have allocated a time budget of 4h per paper, and my review is based on that. In particular I did not read the supplementary material in greater detail. I regret this situation and apologize for possible inaccuracies.

**Strengths:**

- A framework to interpret attention mechanisms in Transformers.
- Principled design of new types of attention is possible by reasoning in the proposed framework.
- Original contribution to literature linking iterative attention to variational inference.



**Weaknesses:**

- The proposed multi-hop and expanding attention designs are only evaluated on toy tasks with structure matching the design.
- Significant discussion and additional results are hidden in the supplementary material, and the presentation in the main text is dense at times. It might seem that the paper would be more coherent and better to understand in a journal/venue with larger page limit.


**Questions:**

- The equation showing the attention definition in the introduction is not directly referenced or explained. Also $d_k$ is undefined.
- Since cross attention and self attention only differ by setting $x'=x$ for self attention, sections 3.2 and 3.3 could easily be merged, resulting in additional space being available for larger Figures 2 and 3 (which are essentially unreadable on A4), or for additional explanations in the beginning to not loose non-expert readers: For example clarifying why $p(\phi|x)=softmax(\ln p(x,\phi))$ in line 97/98, or making the paragraph on pMRFs in lines 109-115 more accessible.
- Would introducing temperature scaling in the softmax lead to a useful generalization?
- In the expression for $F(x,\mu)$ in line 177, the normalization $Z$ is missing (although of course it drops out in $\partial_{\mu}F$).
- How eq. (7) follows from $\partial_{\mu}F = 0$ is not immediately clear.
- In sect. 4.2, it was not clear to me why $z_i$ is can be replaced by $\mu_i$ in the eq. following line 192. Also, is there an asterisk missing at the $\mu_i$ on the rhs here and in eq. (7)?
- In sect. 4.4, how does the uniform prior over incoming synapses relate to the standard version of predictive coding?
- Is the task described in sect. 5.1 lines 243-249 motivated as a toy version of a real world task?
- In sect. 5.2, the description currently does not reference the supplementary material.


Some typos: \
 l109: comma missing after partition function \
 l159: Laplace \
 l193: the system the fixed point \
 l261: the size of the size of \
 l262: "the" dimension \
 l292: . instead of ,

**Limitations:**

A short but reasonable discussion of limitations is provided.

---

> ### Author Rebuttal · Authors · 2023-08-09
>
> Thank you for your review,
>
> - *Since cross attention and self attention only differ by setting $x=x'$ for self attention, sections 3.2 and 3.3 could easily be merged*
>
> In hindsight we agree with this, and would reduce self-attention and cross-attention to a single section.
>
> - *Would introducing temperature scaling in the softmax lead to a useful generalization?*
>
> Within the framework there are two ways of accommodating temperature, either as a parameter of the graphical model (in the edge potentials) or in terms of  *tempered/safe* Bayes. Indeed including the temperature as a latent variable represents an interesting direction, however for space reasons we felt it would confuse the presentation here.
>
> - *How eq. (7) follows from  is not immediately clear.*
>
> This follows from application of the Convex-Concave Procedure (instead of gradient descent), giving a fixed point equation which necessarily reduces the objective function. We would be happy to make this clearer in a revision.
>
> - *In sect. 4.2, it was not clear to me why $z_i$ is can be replaced by $\mu_i$ in the eq. following line 192.*
>
> This is due to the Laplace approximation where we are updating the variational parameters rather than the unknown variable z, however we could discuss this more explicitly for increased clarity.
>
> - *Also, is there an asterisk missing at the $\mu_i$ on the rhs here and in eq. (7)?*
>
> We use the asterisk on the L.H.S to indicate an approximate stationary point (solution), whereas the right hand side is determined by initialization and not necessarily a fixed point. We now realise this is not mentioned, so thanks for pointing it out, again we could make this clearer in the final presentation.
>
> - *how does the uniform prior over incoming synapses relate to the standard version of predictive coding?*
>
> In standard formulations of predictive coding the equivalent term is governed by a precision matrix (of the generative model), learnt slowly across the data, however in practice this term is fixed as a scaled identity matrix and so acts as a uniform weighting over incoming signals (cf. uniform prior).
>
> Additionally, even when they are learnt these matrices are treated as parameters, rather than latent variables, and so remain fixed during inference (no in-context updates) arguably making the structural inference perspective a more powerful model of attention.
>
> - *Is the task described in sect. 5.1 lines 243-249 motivated as a toy version of a real world task?*
>
> Since this concern was brought up more than once we have included the response in the global comment.

---

> > ### Comment · Reviewer_14Rb · 2023-08-15
> >
> > Thank you for your response. I do not have additional questions at the moment, and keep my high score also in the light of the other reviews.

---

### Official Review · Reviewer_JZQo · 2023-07-03

**Soundness:** 2 fair
**Presentation:** 1 poor
**Contribution:** 2 fair
**Rating:** 5
**Confidence:** 1

**Summary:**

This work presents a theoretical framework of how the attention mechanism often used in transformers can be recast as inference over possible adjacency structures in graphical models. In particular, there is an implicit inference on the distribution of edges within a graphical model defined over nodes in the query and key nodes. They can explain many kinds of recent models in this framework such as slot attention, modern hopfield nets, etc. Then, they do two small experiments illustrating the use in using the framing. I am not familiar enough with the literature to know how novel this framing is.

**Strengths:**

* The methods seem very theoretically rigorous.

* Lots of connections to other models in the literature.

**Weaknesses:**

* For the first toy problem, there could be more description as to why having a two-hop neighborhood would be advantageous, or what kind of data would have this property. The second toy problem's motivation was much more clear to me.

**Questions:**

* The analysis to Predictive Coding Networks is interesting. It would be nice to know how neuroscience models of hippocampal structural inference can be explained under this framework (Clone Structured Cognitive Graphs, Tolman Eichenbaum Machine, etc).

**Limitations:**

Limitations were addressed in section 6.1.

---

> ### Author Rebuttal · Authors · 2023-08-09
>
> Thank you for your review,
>
> - *For the first toy problem, there could be more description as to why having a two-hop neighborhood would be advantageous, or what kind of data would have this property. The second toy problem's motivation was much more clear to me.*
>
> Since this concern was brought up by more than one reviewer we have addressed it in the global comment.
>
> - *The analysis to Predictive Coding Networks is interesting. It would be nice to know how neuroscience models of hippocampal structural inference can be explained under this framework (Clone Structured Cognitive Graphs, Tolman Eichenbaum Machine, etc).*
>
> Thank you, we agree that the relationship to hippocampal function and indeed the transformer-TEM is an interesting direction that we are excited about pursuing. For now, it remains unclear exactly what the relationship is, other than it may be important for the hippocampus to perform discrete (relational) inferences in-context (cf. structural inference) while more abstract general properties are consolidated in cortex (i.e. the complementary learning systems hypothesis).

---

> > ### Comment · Reviewer_JZQo · 2023-08-14
> >
> > Thanks for the response. I will keep my score as is.

---

### Official Review · Reviewer_khyL · 2023-07-04

**Soundness:** 3 good
**Presentation:** 3 good
**Contribution:** 2 fair
**Rating:** 6
**Confidence:** 3

**Summary:**

The paper proposes a framework for interpreting standard formulations of attention mechanisms through the lens of graphical models. The authors illustrate that their formulation unifies architectures and offers a way to easily generalize and improve the existing formulations.

**Strengths:**

- The paper offers an interesting direction of looking at existing formulations from the lens of probabilistic models, which should open up several exciting works in the future.
- Interpreting the attention mechanism in terms of an implicit probabilistic model opens up the possibility of understanding & changing the underlying modeling assumptions in a principled manner to design new attention mechanisms.
- The paper is well-written and easy to follow.
- In addition, the authors also include quantitative analysis to demonstrate that their framework can assimilate the table structures to generate robust representations.


**Weaknesses:**

One of the main issues in the paper lies in the experimental evaluation & baselines.

I encourage the authors to include a more rigorous evaluation of their approach, comparing the proposed framework with the formulations of stochastic attention in the literature. The authors chose to demonstrate the results on toy problems, and it is hard to evaluate the effectiveness of the approach based on those results.

Without appropriate experiments, it is hard to judge the utility of the proposed approach; though intuitively, it could do well, I anticipate issues in practice.

How scalable are the proposed changes to the attention mechanisms? How do these changes interplay with popular architectures?

Does the formulation impose any constraints on decoding algorithms? For instance, does it affect the scalability & run time?

In addition, a few relevant references need to be included. Please refer to the list below:
Shujian Zhang, Xinjie Fan, Bo Chen, Mingyuan Zhou: Bayesian Attention Belief Networks. ICML 2021
Hareesh Bahuleyan, Lili Mou, Olga Vechtomova, Pascal Poupart: Variational Attention for Sequence-to-Sequence Models. CoRR abs/1712.08207 (2017)
Shiv Shankar, Sunita Sarawagi Posterior Attention Models for Sequence to Sequence Learning  In ICLR, 2019.

**Questions:**

Refer to Weaknesses

**Limitations:**


The authors do include a section on limitations, but I would say it isn't complete. I encourage the authors to refer to the questions & address them in the list of limitations.

---

> ### Author Rebuttal · Authors · 2023-08-09
>
> - *How scalable are the proposed changes to the attention mechanisms? How do these changes interplay with popular architectures?*
>
> While we see no reason, in principle, for scaling issues, since both modifications were designed with computational complexity in mind. (Multihop requires a single extra matrix multiplication which is cheap compared two two-layers, while expanding is specifically designed to minimise the amount of computation due to context window length.) It remains to be seen if the distributional properties exploited in the toy examples are present in natural data.
>
> More broadly,  we see our key contribution as a theoretical framework which can be used to develop or reason about architectures, while our small experiments serve as proof of principle that this approach works. We would be happy to make this more explicit in the limitations section of the paper.
>
> - *Does the formulation impose any constraints on decoding algorithms? For instance, does it affect the scalability & run time?*
>
> We are not quite sure what is meant here by decoding algorithms, but would be happy to discuss further.
>
> - *..comparing the proposed framework with the formulations of stochastic attention in the literature*
> - *In addition, a few relevant references need to be included*
>
> Thank you, we will certainly include these references in the paper. Although we appear to have missed these, we would like to point out we did highlight the connection to stochastic attention (alignment) formulations (L.59-72) that we feel does capture the main approaches of prior work. Indeed for *Bayesian Attention Belief Networks* and *Posterior Attention Models for Sequence to Sequence Learning* we cited earlier contributions from (a subset of) the authors containing the key ideas.

---

> > ### Comment · Reviewer_khyL · 2023-08-17
> >
> > Thank you for response, I appreciate it, based on the rebuttal, and other reviews, I have decided to increase my score.
> >
> > "Does the formulation impose any constraints on decoding algorithms? For instance, does it affect the scalability & run time?" I was curious if the the modified attention mechanism improves decoding, i.e. produces better outputs, in addition, I also wanted to understand if interpreting attention mechanisms as graphical models would enable us to design better decoders?

---

> > > ### Author Response · Authors · 2023-08-21
> > >
> > > In terms of decoding quality, we imagine a mechanism like multi-hop attention could reduce the number of parameters dedicated to approximating some functions thus freeing up parameters for other tasks. While having a context-size that scales in a data-dependent manner should enable better performance on tasks with long range dependencies in the sequences. However, in both of these cases we imagine both of our design suggestions can be improved upon, here we wished to show that there at least exists some data where they would perform better (synthetic data).
> > >
> > > In general, we believe interpreting the mechanisms through graphical models will definitely enable better design. For example including more structured prior distributions over the edges, or placing priors over hyper-edges to leverage higher order correlations. Aside from changing the model, approximate inference techniques from the graphical model literature could be leveraged to reduce the computational overhead of attention.

---

### Official Review · Reviewer_psYk · 2023-07-05

**Soundness:** 3 good
**Presentation:** 3 good
**Contribution:** 3 good
**Rating:** 7
**Confidence:** 4

**Summary:**

This paper proposes a probabilistic interpretation of attention mechanism, where the computation of attention can be expressed as the expectation of a value function defined on the nodes of a graph consisting of the query nodes and key nodes, and the expectation is taken with respect to the posterior distribution over the edges of the graph. Under such interpretation, different "soft" attention mechanisms fall under the same probabilistic framework, with different node configurations and prior distribution over the edges. Building upon existing works of linking attention mechanisms with variational learning with Gaussian mixture model, the authors additionally established the link between variational inference over the edges (graph structure) and slot attention and hopfield networks. Based on the probabilistic insights, the authors propose new modelling assumptions for designing new attention mechanism.

**Strengths:**

- The proposed probabilistic interpretation for attention mechanisms is sounding and widely applicable to various attention types;
- The new perspective on the link between attention and predictive coding network based on the probabilistic framework is interesting;
- The evaluation of the proposed new attention mechanisms on toy problems exhibit promising results;

**Weaknesses:**

- Despite the mathematical derivations are sounding, the paper can be hard to read since the connection based on the graph formulation is implicit, arguably this can be omitted due to page limits, but the authors should consider improving on this respect;

**Questions:**

- d is not defined on line 51;
- Figure 2 and 3 can be made larger for readability;

**Limitations:**

Yes

---

> ### Author Rebuttal · Authors · 2023-08-09
>
> Thank you for your review,
>
> since both your questions were raised by more than one reviewer we have included the answer in the global response.

---

### Official Review · Reviewer_tiWP · 2023-07-06

**Soundness:** 3 good
**Presentation:** 1 poor
**Contribution:** 2 fair
**Rating:** 5
**Confidence:** 3

**Summary:**

This submission describes how many different transformer architecture variants can be see as implicit structural inference.  The inference is understood as taking an expectation over possible connectivity structures constrained by a prior over structures.  Several variants of attention are shown to be describable in this framework (with more in the appendix).  Two new 'designs' of attentional systems are described and illustrated with two briefly-described experiments.

**Strengths:**

The idea of unifying related architectures under a common Bayesian inference framework seems useful and the authors have thoughtfully engaged with a number of related architectures to characterize how they can be integrated under a common framework.

The idea of allowing context to select a graph over which to perform inference is important and interesting and I would like to see it developed further.

The experiments introduce ways of thinking about how transformer style architectures could be extended in novel ways.



**Weaknesses:**

The paper is largely limited to describing various existing models within the attention as expectation over structures framework.  The new experiments are described extremely briefly and the neural network architecture used was very hard to discern, even after a close reading of the appendix.

The idea of allowing context to select a graph over which to perform inference was not developed.  I thought at first that the expanding attention experiment was going to address this, but, if I understand correctly, there are several different experiments each with a fixed p, so that there is no run-time context sensitivity.

**Questions:**

A more selective presentation of the core ideas (e.g. focusing on 2-3 examples rather than 5 with the others relegated to the appendix, coupled with a more extensive set of investigations addressing the above points would have resulted in a more useful and impactful contribution.

I thought the paper would have benefited by showing how a graph was actually latently inferred by a learning experiment -- the title suggested that such an idea might be forthcoming, but it didn't seem to be.  If the attention allocation illustrated in the lower right of Fig 2 was intended to demonstrate what the authors meant by this, they should have made this more explicit.  But if this is what they meant, perhaps the finding is disappointing.  Indeed, the failure of one layer of one-hop attention to learn the problem in 5.1 suggests that any interesting graph that is learned by a transformer must depend on multiple layers and/or heads.

The authors have a whole page in the main text in which they could have developed their presentation more fully.  A revision might use this space to address some of these questions or expand the presentation and analysis of the experiments.



**Limitations:**

The limitations section of the paper does hint at some of the weaknesses mentioned and points to possible future directions.  There are no ethical concerns with this research.

---

> ### Author Rebuttal · Authors · 2023-08-09
>
> Thank you for your thoughtful review.
>
> - *The paper is largely limited to describing various existing models within the attention as expectation over structures framework.*
>
> We see our key contribution as a unifying theoretical framework helping to understand the fundamental computation underlying attention, which is why we thought it appropriate to dedicate so much space to recovering models.
>
> -  *The new experiments are described extremely briefly and the neural network architecture used was very hard to discern, even after a close reading of the appendix.*
>
> We apologise for this, given a revision we could make the architectures clear through inclusion of pseudocode.
>
> - *The idea of allowing context to select a graph over which to perform inference was not developed. I thought at first that the expanding attention experiment was going to address this, but, if I understand correctly, there are several different experiments each with a fixed p, so that there is no run-time context sensitivity.*
>
> Indeed we see the key property unifying the different attention mechanisms is their use of context to determine a graph over which to perform inference at runtime.  We believe the theory we developed here describes exactly this.
>
> While the experiments here were not designed specifically to demonstrate this property (rather to use our understanding to design extensions to the attention mechanism) they also exhibit inference time context sensitivity.
>
> Specifically, in the expanding task with different p; the p value is a parameter for the data generating (task) distribution*, not the model —- the model determined the context window based on an individual instance of the task, growing the window as needed (on a per data-point basis).
>
> *The parameter is of a geometric distribution, i.i.d. draws from which determine the distance of the signal token from final token. This distance is therefore still different per datapoint.
>
> - *Indeed, the failure of one layer of one-hop attention to learn the problem in 5.1 suggests that any interesting graph that is learned by a transformer must depend on multiple layers and/or heads*
>
> We agree multiple heads or layers are crucial for learning complex functions, however we believe interesting graphs can be inferred by a single head.
>
> The reason this is not evident in 5.1, the one-hop attention used a restricted internal dimension (keys, queries)  (supplementary L.126)  where typically the embedding space is large enough to incorporate multiple types of relationship within a single head leading to more interesting graphs.

---

### Author Rebuttal · Authors · 2023-08-09

Thank you to the reviewers for their insightful comments. A couple of concerns were repeated across reviewers which we will address here.

We appreciate reviewers concern that the experiments were on toy data, however, we would like to stress that we view the main contribution of the paper as theoretical, providing a perspective on why attention mechanisms are so useful, and unifying different uses of the attention mechanism. While the experiments serve as a proof of principle that we can use this understanding to design new mechanisms, hopefully enabling future researchers to develop new architectures which scale to natural data.

- *What is a natural motivation for the task set-up in multihop attention?*

Requiring two previous states in order to generate the next one serves as a natural example. Consider a character-level sequence model for English language; attending to the letters “ee”, since they are almost always followed by a consonant,  greatly reduces the uncertainty of the next character compared to simply attending to a single “e”. Of course, such cases can be handled with higher capacity heads, multiple heads or multiple layers. However it is possible that using some multi-hop heads could aid learning efficiency if patterns such as this are common in the dataset.

More generally, if the sequence has useful information spread across two tokens, that is not exclusively available from either of them individually, multihop could serve as an alternative to increasing the number of parameters (e.g. capacity, layers or heads).

- *Figure size too small*

We will make sure we increase the figure sizes in future versions.

-  *$d$ not defined in first equation.*

Thanks to reviewers for pointing this out, we will make sure this is defined.

---

> ### Comment · Reviewer_tiWP · 2023-08-12
> **Responses somewhat helpful**
>
> I appreciate that the authors responded to my comments and those of the other reviewers, and I now understand the p parameter better in the response to my comments, but my overall impression of the extent of the contribution here has not changed.  I strongly encourage the authors to pursue the research further and develop a presentation that brings out the important points more fully.

---

### Comment · Area_Chair_2P5M · 2023-08-18
**Inquiry regarding MLPs**

Dear Authors,

In transformer models, self-attention is followed by an MLP layer. This work focuses solely on the attention layer. I am wondering whether intuitions extend to the combined design of attention and MLPs. More generally, is there any implication of this work when attention is followed by downstream nonlinearities?

Thank you,

Area Chair

---

> ### Author Response · Authors · 2023-08-19
> **On the role of the MLP**
>
> Although we did not consider the MLP (ffn) here, we do think this perspective can provide some insights into it’s function, however these are necessarily of a more speculative nature. We will focus on two distinct claims on the role of the ffn:
>
> 1. **The ffn acts as a key-value memory store** [1, 2]
> 2. **The ffn prevents representation collapse**  [3, 4, 5]
>
> and offer a potential route to see these as two sides of the same coin.
>
> 1. **The ffn acts as a key-value memory store**
>
>  This observation was inspired by noting the similarity of $ffn(x)= \sigma(W_1x)W_2$ (omitting biases) to cross attention $CrAtt(q)= softmax(Kq)V$.  Which we know is equivalent to the MCHN (in fact more similar since the $K$ memories are stored parameters rather than transient activations).
>
> As we have shown here this memory component can also be thought of as structural inference (where the edge connecting each token to a memory is inferred in context). Aside from being able to write the whole transformer block as interpretable operations on a single probabilistic model, we could also motivate alternative updating schemes using the iterative inference framework (similar to block-slot attention in supplementary):
> $$
> y^*=\sum_{\phi_x}p(\phi_x \mid x)\nabla_y \ln p(y\mid \phi_x, x) +\sum_{\phi_m} p(\phi_m\mid w)\nabla_y \ln p(y \mid \phi_m, w)
> $$
> which is naturally parameterised with learnable components as $
> y_{new}=Attn(x) + ffn(y_{old})
> $. When $y_{old}$ is initially set to $x$  this update resembles the parallel updates proposed here [5], however the authors motivation was with respect to the second role of the ffn.
>
> 2. **The ffn prevents representation collapse**
>
> A different way of thinking about feedforward components is as preventing “token collapse”. It has been shown that removing either skip or feedforward components of transformers increases the rate of tokens collapsing to a single representation [3, 4, 5]. Intuitively, this is related to diffusion processes on connected graphs - after some amount of time all signals become well mixed. The same phenomena has caused issues in training deep graph neural networks [6].
>
> Our framework provides a natural bridge to graph neural networks, and also suggests alternative methods to prevent oversmoothing. For example, placing zero priors on fully-connected graphs (hence zero posterior) could be important, especially in later layers. Alternatively depth dependent modifications to the temperature (c.f. tempered bayes, effectively controlling the strength of the uniform prior) as suggested by [4].
>
> 3. ************************Potential unification between 1. and 2.************************
>
> Multiple studies have shown (by early-decoding) that multiple layers are actually acting as iterative refinements [8]. This indicates we might be able to think of the successive layers as roughly implementing an iterative attention mechanism:
> $$ \mu^{t+1} =  \mu^{t} + \lambda_t  E_{\phi \mid \mu_{t}}[\nabla_{\mu} ELBO(\mu^{t}, \phi)]$$
> Note here the similarity to the computation of a residual $x_l = x_{l-1} + F_l(x_{l-1})$, previous work has explored the role of emerging iterative inference in ResNets [7] and indeed this notion of learning to infer is similar to *iterative amortised inference* [9]). This also suggests a perspective on why scaling the residual connections can prevent mixing, by bringing them closer to implicit gradient updates. Taking this view we can see each layer as a successive approximation to the posterior for the $N$ tokens.
> $$
> p(y_{1:N} \mid x_{1:N}) \propto p(x_{1:N}\mid y_{1:N
> })p(y_{1:N}; m)
> $$
> Where the likelihood component corresponds to the attention mechanism and the ffn relates to the prior. We speculate simply maximising the likelihood here is a bad objective and can be solved with $y_{i}=A\bar{x}$   $\forall i$, however the prior, informed by memory acts as a regulariser preventing collapse.
>
> While the ideas here clearly require further investigation and validation, we believe this framework does offer initial insight on the feedforward component and would be excited about pursuing this direction in future work.

---

> > ### Author Response · Authors · 2023-08-19
> > **On the role of the MLP (References)**
> >
> > 1] Geva, M., Schuster, R., Berant, J., & Levy, O. (2021). *Transformer Feed-Forward Layers Are Key-Value Memories* (arXiv:2012.14913)
> >
> > [2] Sukhbaatar, S., Grave, E., Lample, G., Jegou, H., & Joulin, A. (2019). *Augmenting Self-attention with Persistent Memory* (arXiv:1907.01470)
> >
> > [3] Dong, Y., Cordonnier, J.-B., & Loukas, A. (2021). Attention is not all you need: Pure attention loses rank doubly exponentially with depth. *Proceedings of the 38th International Conference on Machine Learning*, 2793–2803
> >
> > [4] Noci, L., Anagnostidis, S., Biggio, L., Orvieto, A., Singh, S. P., & Lucchi, A. (2022). Signal Propagation in Transformers: Theoretical Perspectives and the Role of Rank Collapse. *Advances in Neural Information Processing Systems*, *35*, 27198–27211.
> >
> > [5] Sonkar, S., & Baraniuk, R. G. (2023). *Investigating the Role of Feed-Forward Networks in Transformers Using Parallel Attention and Feed-Forward Net Design* (arXiv:2305.13297)
> >
> > [6] Rusch, T. K., Bronstein, M. M., & Mishra, S. (2023). *A Survey on Oversmoothing in Graph Neural Networks* (arXiv:2303.10993).
> >
> > [7] Jastrzębski, S., Arpit, D., Ballas, N., Verma, V., Che, T., & Bengio, Y. (2018). *Residual Connections Encourage Iterative Inference* (arXiv:1710.04773).
> >
> > [8] Belrose, N., Furman, Z., Smith, L., Halawi, D., Ostrovsky, I., McKinney, L., Biderman, S., & Steinhardt, J. (2023). *Eliciting Latent Predictions from Transformers with the Tuned Lens* (arXiv:2303.08112).
> >
> > [9] Marino, J., Yue, Y., & Mandt, S. (2018). Iterative Amortized Inference. *Proceedings of the 35th International Conference on Machine Learning*, 3403–3412.

---

### Decision · Program_Chairs · 2023-09-21

**Decision:**

Accept (poster)

**Comment:**

I recommend acceptance of the paper primarily due to its novel and theoretically-grounded approach to interpreting attention mechanisms through a probabilistic framework. The probabilistic viewpoint introduced in this work can have wider applications and motivate promising avenues for future research. The reviewers also found presentation to be clear and results to be novel, even though they are based on toy problems. However, there is room for improvement in experimental validation and in providing clearer motivations for the choice of toy problems. Despite these limitations, the reviewers and I believe the work's contributions are significant enough to warrant acceptance. For the final version, I recommend that the authors revise their work based on reviewer feedback. I personally would also like to see a discussion on MLPs/nonlinearities following attention but this is certainly at the authors' discretion.